# Meta-Learned Surrogates for Clustering Model Selection

## Abstract

Clustering model selection without ground-truth labels relies on Internal Validity Measures (IVMs) such as Silhouette, Calinski-Harabasz, and Davies-Bouldin. These fixed surrogates encode particular geometric assumptions and often correlate poorly with external agreement across heterogeneous datasets. We propose METAIVM, a meta-learned surrogate for external-agreement-based clustering model selection. Trained offline on labeled benchmarks and deployed without labels, METAIVM predicts the quality of individual (dataset, partition) pairs from observable features of partition structure, dataset statistics, and graph topology. Unlike prior meta-learning work that recommends algorithms at the dataset level, this per-partition formulation handles algorithm choice, hyperparameter selection, and cluster-count selection in a single framework. On a benchmark of 223 datasets and 16,889 clustering runs, METAIVM reduces selection regret by 67% over the best classical IVM. Principled controls show that neither learning from IVMs alone nor dataset-level meta-selection suffices: the per-partition formulation is essential, and even linear regression with our features outperforms all IVMs. The method adapts its feature reliance to dataset geometry, is robust across four external metrics and across modified candidate pools, and transfers from synthetic to real-world domains, though performance depends on the training distribution. As a preliminary extension to graph community detection, where coordinate-based IVMs do not apply, METAIVM outperforms modularity-based selection.

## 1 Introduction

Clustering is one of the most widely used unsupervised learning paradigms, with applications spanning biology, finance, computer vision, and natural language processing. A fundamental challenge in clustering is *model selection*: given a dataset and a set of candidate algorithms with varying hyperparameters, how does one select the best clustering without access to ground-truth labels?

The standard approach relies on *Internal Validity Measures* (IVMs), scores computed from the data and the partition alone, to rank candidate clusterings. Popular IVMs include the Silhouette Coefficient (Rousseeuw, 1987), Calinski-Harabasz Index (Caliński & Harabasz, 1974), and Davies-Bouldin Index (Davies & Bouldin, 1979). The implicit assumption is that these measures correlate with external agreement across datasets, so maximizing them leads to good model selection.

In practice, this assumption often fails. Prior work has shown that IVMs exhibit highly inconsistent behavior across datasets: a measure that correctly identifies the best clustering on one dataset may actively prefer the worst clustering on another (Simpson et al., 2026; Jeon et al., 2025). In our experiments across 223 datasets, the best classical IVM (Calinski-Harabasz) achieves only $\rho = 0.395$ average per-dataset Spearman correlation with external agreement (AMI), and a selection regret of 0.217, meaning it selects clusterings that are, on average, 21.7% worse than optimal. The Silhouette Coefficient performs even worse ($\rho = 0.062$, regret $= 0.342$).

We propose to treat clustering model selection as a *surrogate prediction problem*: rather than relying on hand-crafted validity measures, we learn a surrogate for external clustering agreement from data. Our method, METAIVM, is trained on datasets with known ground-truth labels and deployed without labels. Given a new (dataset, partition) pair, it predicts external agreement (operationalized primarily through Adjusted Mutual

Information, AMI) between the partition and the unavailable ground truth, using features derived from the partition structure, dataset statistics, and graph-based measures. This *per-partition* formulation is a key distinction from prior meta-learning work on clustering, which operates at the dataset level (recommending an algorithm family) rather than scoring individual candidate partitions. By predicting quality for specific (dataset, partition) pairs, MetaIVM naturally handles algorithm choice, hyperparameter selection, and cluster-count selection in a single framework. The framework is model-agnostic: Ridge, MLP, and XGBoost on the same features all substantially outperform classical IVMs.

The key insight is that the evidence predictive of external agreement is context-dependent, varying with data geometry. On spherical clusters, dispersion ratio is the strongest signal; on manifolds, graph conductance dominates; on imbalanced data, cluster size statistics carry most of the information. No single fixed IVM can adapt its evaluation criterion (Appendix J provides a formal illustration for compactness-separation IVMs). A supervised model trained on diverse (dataset, partition, quality) triples learns which signals matter where. Ablations support that the main gains come from the formulation and feature space rather than a specific model: removing IVM features entirely barely affects performance (regret 0.067 vs. 0.065 with all features), and even linear regression beats all IVM baselines (regret 0.088 vs. 0.217).

Our contributions are:

1. **Per-partition quality prediction.** We formalize clustering model selection as predicting external agreement for individual (dataset, partition) pairs, a more general task than prior meta-learning work that selects algorithms at the dataset level. Dataset-level train/test splits prevent information leakage and enable honest evaluation.

2. **Benchmark and evaluation.** We curate a benchmark of 223 datasets (91 OpenML tabular, 123 synthetic, 5 text, 4 image) with 16,889 clustering runs across 6 algorithm families, evaluated with paired Wilcoxon significance tests and failure-mode characterization.

3. **Large improvements with principled controls.** Our feature set, combined with any standard ML model, substantially outperforms classical IVMs: Ridge achieves 0.088 regret, XGBoost 0.071, a 59–67% reduction over the best IVM (0.217). Learned baselines isolate the source: learning IVMs alone does not help, dataset-level meta-selection improves over CH but remains far from per-partition scoring, and partition-level features are essential. Removing IVM features barely affects performance (regret 0.067).

4. **Adaptive feature reliance.** Per-category analysis shows that MetaIVM shifts its reliance depending on data geometry: variance heterogeneity on ellipsoidal clusters (81%), graph conductance on manifolds (20%), cluster size statistics on imbalanced data (60%), behavior that no fixed IVM can exhibit. The improvement is concentrated on hard datasets where CH fails (83% regret reduction).

5. **Transfer, data efficiency, and robustness.** Synthetic pretraining shows promising transfer to real-world domains (regret 0.100), though transfer is asymmetric. The method outperforms all IVMs with as few as 13 training datasets and remains robust under perturbations to the candidate pool (Section 6). As a preliminary extension, we also apply the same surrogate-learning principle to graph community detection, where coordinate-based IVMs do not apply, obtaining promising results.

## 2 Related Work

**Internal Validity Measures.** Classical IVMs evaluate clustering quality without ground truth by measuring properties such as within-cluster compactness and between-cluster separation. The Silhouette Coefficient (Rousseeuw, 1987) measures how similar each point is to its own cluster versus the nearest neighboring cluster. The Calinski-Harabasz Index (Caliński & Harabasz, 1974) computes the ratio of between-cluster to within-cluster variance. The Davies-Bouldin Index (Davies & Bouldin, 1979) averages the worst-case ratio of within-cluster scatter to between-cluster separation. Despite their widespread use, these measures have known limitations: they are biased toward specific cluster shapes (e.g., convex, spherical), do not generalize well across diverse data distributions, and often disagree with each other (Arbelaitz et al., 2013).

**Density-Based and Specialized IVMs.** Recent work has developed IVMs tailored to specific clustering paradigms. DISCO (Beer et al., 2026) proposes a density-based internal score for clusterings with noise, using density-connectivity distances to handle arbitrary cluster shapes and explicitly evaluating noise label quality, addressing a limitation of classical IVMs that assume all points belong to clusters. While such specialized IVMs improve evaluation within their target paradigm, our approach is orthogonal: rather than designing a better fixed formula for a specific setting, we learn the evaluation function from data across diverse settings.

**Adjusted and Cross-Dataset IVMs.** Jeon et al. (2025) introduced adjusted IVMs ($\text{IVM}_A$) designed to enable fair cross-dataset comparison. Their approach applies a nonlinear shift transformation and normalizes by the expected score under random label permutation, computed via Monte Carlo simulation over all pairwise class comparisons. While $\text{IVM}_A$ addresses the bias of raw IVMs when comparing *across* datasets, we show that for within-dataset model selection, it does not consistently outperform raw Calinski-Harabasz (regret 0.258 vs. 0.217).

**IVM Selection and Combination.** Simpson et al. (2026) studied the instance space of clustering validation measures using 18,351 synthetic datasets and meta-features, showing that different IVMs have complementary strengths and proposing a KNN-based selector to choose the best IVM per dataset. Azevedo et al. (2025) explored combining multiple IVMs via multi-objective optimization with NSGA-II. Our approach differs in that rather than selecting among or combining existing IVMs, we learn a surrogate for external agreement that can incorporate IVMs as features alongside additional structural descriptors, and can also operate without IVM inputs entirely.

**Meta-Learning for Clustering.** Meta-learning has been applied to algorithm selection for clustering (Ferrari & De Castro, 2015; Pimentel & de Carvalho, 2019), where the goal is to predict which *algorithm* will perform best on a given dataset using dataset-level meta-features. Our work addresses a distinct and more general problem: given a specific (dataset, partition) pair, predict the quality of *that partition*. This per-partition formulation subsumes algorithm selection because it naturally handles algorithm choice, hyperparameter selection, and cluster-count selection simultaneously, since any partition produced by any method can be scored. Prior algorithm-selection work cannot differentiate between KMeans with $k=3$ and $k=10$ on the same dataset, because both are "KMeans." We can, because we observe the partition itself. This is analogous to the distinction between algorithm selection and performance prediction in supervised learning (Bischl et al., 2016).

**Graph Community Detection Evaluation.** For graph-structured data, the dominant quality function is modularity (Newman, 2006), which measures the fraction of within-community edges relative to a null model. However, modularity suffers from a well-known resolution limit: it cannot detect communities smaller than a scale that depends on the total number of edges (Fortunato & Barthélemy, 2007). Alternative quality functions include normalized cut, conductance, and surprise (Traag et al., 2015), but no principled method exists for selecting among community detection algorithms on a given graph without ground truth. Our graph extension (Section 6.8) addresses this gap by learning a surrogate quality predictor from graph-topology features.

## 3 Problem Formulation

### 3.1 Setup

Let $\mathcal{D} = \{(\mathbf{X}_i, \mathbf{y}_i)\}_{i=1}^N$ be a collection of datasets, where $\mathbf{X}_i \in \mathbb{R}^{n_i \times d_i}$ is the feature matrix and $\mathbf{y}_i \in \{1, \ldots, k_i\}^{n_i}$ is the ground-truth class label vector. For each dataset $\mathbf{X}_i$, we generate a set of candidate partitions $\{\boldsymbol{\pi}_{ij}\}_{j=1}^{M_i}$ by running multiple clustering algorithms with varying hyperparameters. The quality of each partition is measured by the Adjusted Mutual Information (AMI) between $\boldsymbol{\pi}_{ij}$ and $\mathbf{y}_i$:

$$q_{ij} = \text{AMI}(\boldsymbol{\pi}_{ij}, \mathbf{y}_i) \in [-1, 1]. \tag{1}$$

Our goal is to learn a function $f(\mathbf{X}_i, \boldsymbol{\pi}_{ij}) \to \hat{q}_{ij}$ that predicts external agreement from observable features of the data and the partition, *without* access to $\mathbf{y}_i$ at test time. Crucially, deployment is fully unsupervised: the user runs candidate clustering algorithms (with various hyperparameters and numbers of clusters), and the pre-trained model scores each resulting partition. All features, including the number of clusters, are extracted from the partition output, not specified by the user. This means METAIVM simultaneously addresses both algorithm selection and cluster-count selection, unlike classical IVMs which are known to be biased toward specific values of $k$ (Arbelaitz et al., 2013).

## 3.2 Evaluation: Selection Regret

The primary use case is model selection: given a new dataset $\mathbf{X}^*$ and candidates $\{\boldsymbol{\pi}_j^*\}$, select the partition with highest predicted external agreement:

$$j^* = \arg\max_j f(\mathbf{X}^*, \boldsymbol{\pi}_j^*). \tag{2}$$

We measure performance via *selection regret*:

$$\text{Regret} = \max_j q_j^* - q_{j^*}^*, \tag{3}$$

which is the gap between the external agreement of the best available partition and the external agreement of the partition selected by our method. Lower regret is better; zero means optimal selection. Note that when multiple partitions achieve the same maximum AMI (common on easy datasets where many algorithms find the ground-truth structure), selecting any of them yields zero regret. We complement regret with Spearman $\rho$, which measures ranking quality across *all* partitions, not just the top.

## 3.3 Dataset-Level Splits

To prevent information leakage, we split at the *dataset* level: all partitions from a given dataset appear entirely in train, validation, or test. We use $60/20/20$ splits with 5 random seeds and report mean $\pm$ standard deviation across seeds. This ensures the model is evaluated on datasets it has never seen during training.

# 4 Method

## 4.1 Feature Extraction

For each (dataset, partition) pair, we extract features from four complementary sources:

**Partition statistics** (12 features): Number of clusters, noise fraction, cluster size statistics (min, max, mean, std, median), size ratio, entropy, Gini impurity, imbalance ratio, singleton fraction.

**Dataset descriptors** (14 features): Number of samples and features (and their logs), dimensionality ratio, feature-wise mean/std/skewness/kurtosis (averaged), sparsity, intrinsic dimensionality (PCA 95% variance), and pairwise distance statistics (mean, std, median).

**Partition-data interaction features** (10 features): Within-cluster variance statistics, between-cluster variance, dispersion ratio, cluster density statistics, centroid separation statistics.

**Graph-based features** (6 features): Using a precomputed $k$-nearest-neighbor graph, we compute cut fraction, normalized cut proxy, within-cluster edge density (mean and std), and conductance statistics.

**IVM features** (3 features): Raw Silhouette, Calinski-Harabasz, and (negated) Davies-Bouldin scores are included as features. This allows the model to leverage IVMs as inputs while learning when to trust or override them.

**Algorithm encoding** (11 features, *optional*): One-hot encoding of the algorithm family (6 categories) and numeric hyperparameters (5 values). These are deployment-optional: the ablation (Table 4) shows removing them has minimal impact (regret 0.078 vs. 0.065). Our default configuration excludes them, since they are not always available at deployment time.

In total, we extract 56 features per (dataset, partition) pair. All features are computed without access to ground-truth labels. Our **default deployment configuration** uses the 28 features from the first three groups (partition statistics, partition-data interaction, graph-based), which require only the partition and a precomputed $k$NN graph. Dataset descriptors, IVM scores, and algorithm encoding are additional features evaluated in ablations (Table 4) but excluded from the default model.

## 4.2 Model

METAIVM is model-agnostic: any supervised regression model can be used to map the 56 features to a predicted AMI score. We evaluate three models of increasing complexity:

- **Ridge regression**: $L_2$-regularized linear model. Regularization strength $\alpha$ is selected from $\{0.01, 0.1, 1.0, 10.0, 100.0\}$ via validation set performance, then refitted on train+val with the best $\alpha$.

- **MLP**: Two-layer neural network (256–128 hidden units, ReLU activations, Adam optimizer with $\text{lr} = 10^{-3}$). Trained for up to 500 epochs with early stopping (patience 20) on the validation set. Features are standardized to zero mean and unit variance.

- **XGBoost** (Chen & Guestrin, 2016): Gradient-boosted regression trees. Hyperparameters are selected via grid search over max depth $\in \{3, 5, 7\}$ and learning rate $\in \{0.01, 0.05, 0.1\}$ (9 combinations), each with up to 500 estimators and early stopping (patience 20) on the validation set. The best configuration is refitted on train+val.

All three models substantially outperform classical IVMs (Table 1), supporting the view that the improvement comes primarily from the formulation and feature design rather than a specific model. XGBoost achieves the lowest regret and is used as the default METAIVM model due to its robustness, native handling of missing values, and interpretability via feature importance.

# 5 Experimental Setup

## 5.1 Dataset Collection

We curated a diverse collection of 223 datasets from four sources:

- **OpenML** (91 datasets): Real-world tabular datasets from the OpenML platform (Vanschoren et al., 2014), including the CC18 benchmark suite and additional hand-selected datasets covering biology, finance, physics, and social sciences. Datasets with >20k samples were subsampled; those with >200 features were PCA-reduced to 50 dimensions.

- **Synthetic** (123 datasets): Systematically generated datasets varying dimensionality (2–200), number of clusters (2–10), overlap level, noise fraction, cluster shape (spherical, ellipsoidal, manifold), imbalance ratio, and subspace structure. This ensures coverage of clustering scenarios where different IVMs are known to fail.

- **Text** (5 datasets): 20Newsgroups (Lang, 1995) subsets (all, science, politics, computers, recreation) with TF-IDF features reduced to 50 dimensions via SVD.

- **Image** (4 datasets): MNIST (LeCun et al., 1998) and Fashion-MNIST (Xiao et al., 2017) subsets (2k and 5k samples each) with PCA-50 embeddings.

## 5.2 Clustering Algorithms

For each dataset, we run six algorithm families with varied hyperparameters:

- **KMeans**: $k \in \{2, 3, 5, 7, 10, 15, 20\}$, n_init $= 10$.

- **Gaussian Mixture**: $k \in \{2, 3, 5, 7, 10, 15, 20\}$, covariance $\in \{\text{full}, \text{diag}\}$.

- **DBSCAN**: $\varepsilon \in \{0.3, 0.5, 1.0, 2.0\}$, min_samples $\in \{3, 5, 10\}$.

- **HDBSCAN**: min_cluster_size $\in \{5, 10, 20, 50\}$, min_samples $\in \{3, 5, 10\}$.

- **Agglomerative**: $k \in \{2, 3, 5, 7, 10, 15, 20\}$, linkage $\in \{\text{ward}, \text{complete}, \text{average}\}$.

- **Spectral**: Skipped due to $O(n^3)$ complexity (not scalable to our dataset sizes).

The full hyperparameter grid contains 91 configurations per dataset; after skipping SpectralClustering ($O(n^3)$) and AgglomerativeClustering for datasets with $n > 10,000$ ($O(n^2)$ memory), the effective count ranges from 19 to 79 per dataset (mean: 75.7; lower counts occur for the largest datasets where additional GMM configurations also fail to converge). This yields 16,889 total (dataset, partition) pairs with ground-truth AMI scores.

## 5.3 Baselines

We compare against:

- **Random**: Uniform random selection among candidates.

- **Global best**: The single (algorithm, hyperparameters) configuration with highest average AMI on training data, applied to all test datasets.

- **Classical IVMs**: Silhouette, Calinski-Harabasz, and (negated) Davies-Bouldin used directly for model selection.

- **Adjusted CH** (Jeon et al., 2025): The adjusted Calinski-Harabasz index designed for cross-dataset comparability, computed using the official `btwim` library with 20 Monte Carlo iterations.

- **Ridge regression**: Linear baseline with the same features as XGBoost.

- **MLP**: Two-layer neural network (256–128 hidden units) with the same features.

## 5.4 Metrics

We report five complementary metrics:

- **Selection regret** (primary): Quality gap between the selected and optimal partition.

- **$\varepsilon$-success rate**: Fraction of datasets where regret $< \varepsilon$ (with $\varepsilon = 0.01$).

- **NDCG@5**: Normalized discounted cumulative gain for the top-5 ranked partitions.

- **Spearman $\rho$**: Rank correlation between predicted and true AMI.

- **Kendall $\tau$**: Concordance probability for pairwise comparisons.

Table 1: Model selection performance across 223 datasets (5 seeds, default 28-feature set for learned models). Best in **bold**.

| Method | Regret ↓ | $\varepsilon$-Success ↑ | NDCG@5 ↑ | Spearman $\rho$ ↑ | Kendall $\tau$ ↑ |
|---|---|---|---|---|---|
| Random | $.177 \pm .180$ | $.083 \pm .276$ | $.467 \pm .279$ | $.019 \pm .124$ | $.014 \pm .087$ |
| Global best config | $.241 \pm .219$ | $.113 \pm .317$ | $.554 \pm .346$ | $.058 \pm .098$ | $.049 \pm .082$ |
| Silhouette | $.342 \pm .297$ | $.078 \pm .269$ | $.425 \pm .349$ | $.433 \pm .402$ | $.332 \pm .325$ |
| Calinski-Harabasz | $.217 \pm .237$ | $.187 \pm .391$ | $.607 \pm .311$ | $.604 \pm .320$ | $.471 \pm .283$ |
| Davies-Bouldin | $.345 \pm .305$ | $.096 \pm .295$ | $.405 \pm .362$ | $.399 \pm .385$ | $.298 \pm .313$ |
| CH Adjusted (Jeon et al., 2025) | $.258 \pm .270$ | $.148 \pm .356$ | $.518 \pm .422$ | $.531 \pm .281$ | $.405 \pm .237$ |
| Ridge | $.088 \pm .133$ | $.330 \pm .471$ | $.743 \pm .314$ | $.662 \pm .307$ | $.526 \pm .262$ |
| MLP | $.072 \pm .098$ | $.322 \pm .468$ | $.737 \pm .490$ | $.685 \pm .324$ | $.555 \pm .291$ |
| **MetaIVM (XGBoost)** | $\mathbf{.071 \pm .103}$ | $\mathbf{.365 \pm .483}$ | $\mathbf{.743 \pm .507}$ | $\mathbf{.742 \pm .302}$ | $\mathbf{.612 \pm .275}$ |

# 6 Results

**Note on reported numbers.** We distinguish two model variants throughout the paper:

- **Default (deployment-realistic):** 28 features (partition statistics + partition-data interaction + graph-based), excluding dataset descriptors, IVM scores, and algorithm identity. This requires only the partition and a precomputed $k$NN graph. Regret: **0.071**.

- **Full-feature (upper bound):** All 56 features including IVM scores and algorithm/hyperparameter indicators. Regret: **0.065**. Appropriate when IVM scores are already available.

Unless otherwise noted, all METAIVM results use the default 28-feature configuration. The feature ablation (Table 4) systematically evaluates all subsets. Small numerical differences between tables (e.g., 0.071 in Table 1 vs. 0.076 in Table 18) reflect different feature sets or averaging scopes, as noted in each table caption.

## 6.1 Main Comparison (Experiment 1)

Table 1 presents the main results averaged over 5 dataset-level splits (seeds 0–4). METAIVM (XGBoost) achieves the lowest selection regret across all methods with a regret of $0.071 \pm 0.103$, a 67% reduction compared to the best classical IVM (Calinski-Harabasz, $0.217 \pm 0.237$) and a 72% reduction compared to the adjusted CH of Jeon et al. (2025) ($0.258 \pm 0.270$). The improvement is consistent across all metrics.

Figure 1 shows the full distribution of selection regret across datasets. METAIVM achieves consistently low regret, while IVMs exhibit heavy tails with frequent catastrophic failures (regret $> 0.5$).

A critical observation: even simple linear regression (Ridge) with our features achieves 0.088 regret, already outperforming all classical IVMs (best: 0.217) by a wide margin. This shows that the gains are primarily driven by the meta-learning formulation and feature design. XGBoost provides a further 19% relative improvement over Ridge (0.071 vs. 0.088, $p = 0.022$ by paired Wilcoxon, Table 19) by learning nonlinear feature interactions, but the linear model alone suffices to demonstrate the advantage of our approach over all IVM baselines.

## 6.2 Isolating the Source of Improvement

To determine whether the gain comes from *learning per se*, the *per-partition formulation*, or the *feature space*, we compare against three principled learned baselines (Table 2).

These baselines are designed as controls to isolate the source of improvement, not as exhaustive alternatives. Three findings emerge:

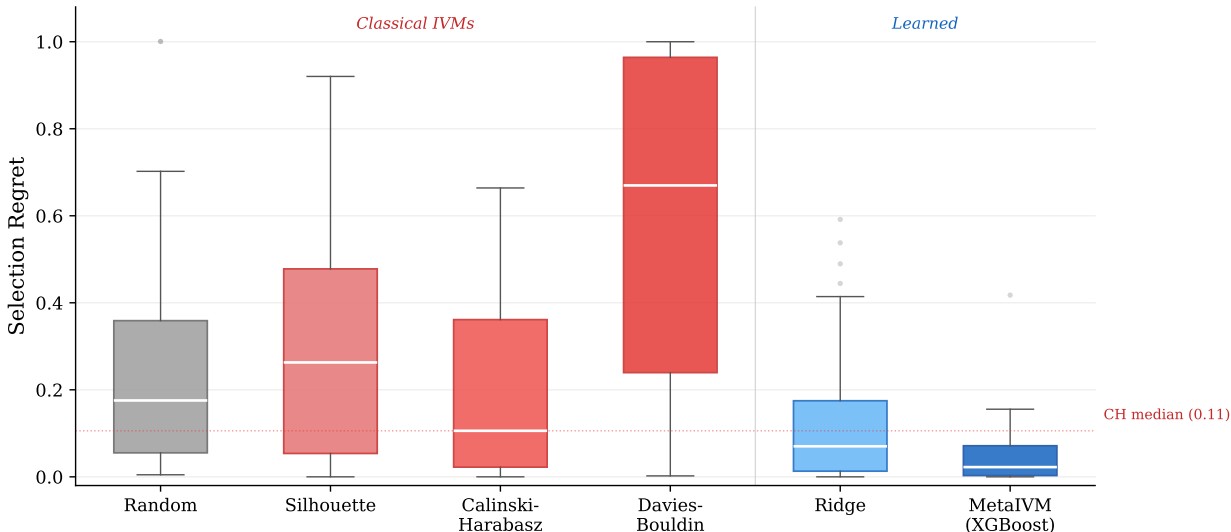

Figure 1: Distribution of selection regret across 223 test datasets using the default 28-feature set (partition statistics + partition-data interaction + graph; no dataset descriptors, IVM, or algorithm features). METAIVM (XGBoost) has the lowest median regret and fewest catastrophic failures. Classical IVMs show high variance with frequent regret > 0.5.

Table 2: Learned baselines isolating the source of improvement (5 seeds). Learning alone (B1) does not help; dataset-level meta-selection (B3) improves over CH but remains far from per-partition scoring; partition-level features are essential.

| Method | $d$ | Regret ↓ | Spearman $\rho$ ↑ |
|---|---|---|---|
| *Fixed IVMs (no learning)* | | | |
| Raw Silhouette | – | .342 | .370 |
| Raw Calinski-Harabasz | – | .217 | .604 |
| *Learned baselines* | | | |
| B1: Ridge on 3 IVMs | 3 | .337 | .391 |
| B2: XGB on IVM + dataset desc. | 17 | .143 | .666 |
| B3: Dataset-level meta-selector[†] | 14 | .180 | – |
| *METAIVM (per-partition features)* | | | |
| Ridge (28 feat, no IVM) | 28 | .123 | .634 |
| **MetaIVM XGB (28 feat, no IVM)** | **28** | **.070** | **.760** |

[†]Best of 5-NN, Ridge, and XGBoost on same descriptors (Ridge selected).

1. **Learning IVMs alone does not help** (B1: regret 0.337 ≈ raw Silhouette). The three classical IVM scores do not contain enough complementary signal for a learned combination to meaningfully improve selection.

2. **Dataset-level meta-selection helps but is insufficient** (B3: 0.180 vs. CH 0.217). The best dataset-level meta-selector (Ridge on 14 dataset descriptors, selected from 5-NN/Ridge/XGBoost via validation) improves over the best fixed IVM baseline, showing that dataset-level learning is useful. However, it remains far worse than per-partition METAIVM (0.070), indicating that dataset-level descriptors alone are insufficient and that observing the candidate partition is critical.

3. **Adding dataset context helps but plateaus** (B2: 0.143). IVM scores combined with dataset descriptors improve over raw IVMs, but without partition-level features the model cannot distinguish

Table 3: Conditional performance by CH difficulty. METAIVM provides the largest gains on hard datasets (CH regret $> 0.3$), winning 13 out of 14 cases.

| Regime | $N$ | CH regret | MetaIVM regret | Gain |
|---|---|---|---|---|
| Easy (CH $\leq 0.05$) | 19 | .015 | .044 | $-.029$ |
| Medium ($0.05 <$ CH $\leq 0.3$) | 13 | .114 | .048 | $+.066$ |
| Hard (CH $> 0.3$) | 14 | .514 | .087 | $+.427$ |
| All | 46 | .195 | .058 | $+.137$ |

between different clusterings of the same dataset. The jump from B2 (0.143) to METAIVM (0.070) comes from observing the partition itself.

**Where does the improvement come from?** The aggregate gains above are concentrated on datasets where classical IVMs fail. Stratifying test datasets by CH difficulty:

On **easy datasets** where CH already succeeds (regret $\leq 0.05$), METAIVM is slightly worse; it over-refines simple cases where the best partition is obvious. On **hard datasets** where CH fails badly (regret $> 0.3$), METAIVM reduces regret from 0.514 to 0.087, an 83% improvement, winning 13 out of 14 cases. This is where the method's value is concentrated: precisely the datasets where classical IVMs provide no useful guidance. A natural follow-up question is whether a hybrid strategy (trusting CH when it is reliable and METAIVM otherwise) could combine their strengths. The challenge is that identifying CH's reliability regime *without labels* is itself the core problem; METAIVM is learning exactly that decision boundary from observable partition and dataset statistics.

### 6.3 Candidate-Pool Robustness

Since regret is defined relative to the candidate pool, a natural concern is whether METAIVM is specialized to the specific menu of algorithms tested. To assess this, we train METAIVM on the full pool and evaluate on systematically modified pools at test time. METAIVM outperforms CH across *all 11 pool configurations tested*, including: dropping each individual algorithm family (regret 0.048–0.059 vs. CH 0.134–0.205), restricting to KMeans-only candidates (0.036 vs. 0.085), restricting to density-based candidates only (0.023 vs. 0.119), random 50% subsampling per dataset (0.055 vs. 0.174), and filtering by cluster count $k \leq 3$ or $k \geq 5$ (0.042–0.054 vs. 0.127–0.139). Notably, the model trained on the full pool generalizes to pools that contain only a single algorithm family at test time. The full breakdown is in Table 23 (Appendix). This supports the view that the learned surrogate captures a general ranking principle over partition quality, not a pool-specific selection rule.

### 6.4 Feature Ablation (Experiment 2)

Table 4 shows the impact of different feature groups. The key findings are:

1. **IVMs as features are useful but not necessary**: Removing all 3 IVM features barely affects performance (regret 0.067 vs. 0.065 for all features). The model without any IVM or algorithm input (42 features, regret 0.075) still dramatically outperforms all classical IVMs (best: 0.217). This directly addresses the concern that METAIVM is "just a learned correction layer on top of CH."

2. **Partition statistics alone are powerful** (regret 0.074, 22 features), outperforming all classical IVMs. Simple features like cluster entropy, Gini impurity, and size ratios carry substantial signal.

3. **Graph features alone** (6 features, regret 0.100) already outperform all IVMs, because $k$NN graph conductance captures cluster separability robustly across shapes.

4. **A compact 15-feature model** (selected as the 15 most important features, stable across $\geq 3$ of 5 training seeds) achieves 0.083 regret, still far better than any IVM. Eight features stable across *all* seeds achieve 0.098 regret.

Table 4: Feature ablation (XGBoost, 5 seeds). $d =$ number of features. "All features" includes IVM + algorithm encoding (56 total). Differences from Table 1 reflect different feature sets.

| Feature Set | $d$ | Regret $\downarrow$ | Spearman $\rho \uparrow$ |
|---|---|---|---|
| Dataset descriptors only | 14 | .303 | – |
| IVM only | 3 | .192 | .603 |
| Graph only | 6 | .100 | .706 |
| Partition only* | 22 | .074 | .755 |
| Partition + dataset | 36 | .078 | .775 |
| **No IVM, no algorithm** | **42** | **.075** | **.794** |
| No IVM features | 53 | .067 | .786 |
| No algorithm features | 45 | .078 | .784 |
| All features | 56 | .065 | .775 |
| Compact: top-15 features | 15 | .083 | .779 |
| Compact: top-8 features | 8 | .098 | .748 |

*Includes partition statistics (12) + partition-data interaction features (10).

## 6.5 Robustness to Algorithm Diversity (Experiment 3)

To test whether the model depends on seeing all algorithm families during training, we perform leave-one-algorithm-out evaluation (Table 14 in Appendix). Removing DBSCAN has the largest impact ($\Delta = +0.023$ regret), consistent with its unique ability to discover non-convex clusters. All other algorithms have minimal impact ($\Delta < 0.007$), indicating robust generalization.

## 6.6 Cross-Domain Transfer (Experiment 6)

Can a model trained on one type of dataset generalize to another? Training exclusively on synthetic point-cloud datasets (blobs, ellipsoids, manifolds) and testing on real-world tabular data yields 0.100 regret vs. CH's 0.235, with $\rho = 0.887$ on unseen image and text datasets (vs. CH's 0.722). Transfer is asymmetric: real→synthetic is weaker (0.238) and underperforms raw CH on synthetic targets (0.193), suggesting synthetic data provides broader geometric coverage but the transfer story is not symmetric. Full breakdown in Table 13 (Appendix).

## 6.7 Additional Analyses

The predicted vs. true AMI scatter (Figure 3, Appendix) shows strong agreement ($R^2 = 0.810$) across the full quality range. MetaIVM achieves per-dataset Spearman $\rho = 0.755 \pm 0.253$, compared to $\rho = 0.395 \pm 0.416$ for the best IVM (Table 15, Appendix). When all 56 features are used, Calinski-Harabasz is the top feature (35% importance), yet removing it barely hurts (Table 4): the model appears to route around CH via redundant structural features when CH is misleading. A compact 15-feature model achieves 0.083 regret, still far better than any IVM. Per-category analysis (Table 5) shows that MetaIVM adapts its reliance across geometries: variance heterogeneity dominates on ellipsoidal clusters (81%), conductance on manifolds (20%), cluster size statistics on imbalanced data (60%). This adaptive behavior, which no fixed IVM can exhibit, is consistent with the observed performance gap.

The improvement over all classical IVMs is highly significant (paired Wilcoxon $p < 10^{-12}$; Table 19, Appendix). The method is robust across four external metrics: AMI, ARI, NMI, and V-measure all show consistent rankings (Table 18, Appendix). With just 13 training datasets (10%), MetaIVM already achieves 0.116 regret, better than all IVMs (Table 20, Appendix). Zero-engineering approaches (graph spectral embeddings, GNN) also outperform all IVMs (Appendix G), suggesting the formulation is robust to feature design choices.

Table 5: Dominant features shift across dataset geometries; the model adapts its reliance to data structure.

| Dataset type | Dominant feature (importance) | Why it matters |
|---|---|---|
| Spherical clusters | Entropy (0.54) | Balanced cluster sizes signal correct $k$ |
| Ellipsoidal clusters | Within-var std (0.81) | Heterogeneous variance reveals shape mismatch |
| Manifold clusters | Conductance (0.20) | Graph structure captures non-convex geometry |
| Imbalanced clusters | Cluster size std (0.60) | Correct partitions preserve the imbalance pattern |
| Real-world data | Entropy (0.17), centroid sep (0.11) | No single signal dominates |

Table 6: Community detection model selection on 80 benchmark graphs (2,000 runs, 5 seeds). Classical IVMs (Silhouette, CH, DB) cannot be computed because no point coordinates exist. METAIVM outperforms all graph-based heuristics.

| Method | Regret ↓ | Spearman $\rho$ ↑ |
|---|---|---|
| *Not applicable (require point coordinates)* | | |
| Silhouette / CH / DB | n/a | n/a |
| *Graph-based heuristics* | | |
| Random | $.259 \pm .318$ | – |
| Coverage | $.677 \pm .341$ | $-.259 \pm .489$ |
| Normalized Cut | $.677 \pm .341$ | $-.259 \pm .489$ |
| Conductance | $.655 \pm .357$ | $-.226 \pm .506$ |
| Modularity | $.045 \pm .088$ | $.872 \pm .205$ |
| **MetaIVM (graph features)** | $\mathbf{.018 \pm .044}$ | $\mathbf{.946 \pm .107}$ |

Failure analysis (Table 21, Appendix) reveals that the model struggles on datasets with high noise, subspace structure, or extreme imbalance; 19% of test datasets have regret > 0.15. Text and image domains have zero high-regret cases.

### 6.8 Extension: Graph Community Detection

Classical IVMs require point coordinates and **cannot be applied** to graph community detection. METAIVM is naturally applicable because our graph-only features (conductance, edge density, normalized cut proxy) require only the adjacency structure and a partition. We evaluate on 80 benchmark graphs (SBM, 2–10 communities, 5 algorithms at multiple resolutions; Table 6).

METAIVM achieves 0.018 regret ($\rho = 0.946$), a 61% reduction over modularity, while coverage, normalized cut, and conductance are all anti-correlated with external agreement ($\rho < 0$). On 7 real-world attributed graphs (Table 16, Appendix), METAIVM achieves 0.004 regret with just 6 training graphs, 86% better than modularity. Together, these experiments demonstrate that the surrogate-learning principle extends to settings where classical IVMs cannot operate.

## 7 Discussion

**Why do IVMs fail, and why does MetaIVM succeed?** Classical IVMs measure fixed geometric properties that correlate with external agreement only when clusters match geometric assumptions. Our analysis (Table 5) reveals that the predictive signal shifts across regimes, and no fixed IVM can adapt. Ablations support that gains come from the formulation and feature space: Ridge alone beats all IVMs, removing IVM features barely hurts, and learned baselines (Table 2) show neither learning alone nor dataset-level selection suffice. The model uses CH when informative and routes around it when misleading.

**Scope and assumptions.**    Our goal is not universal clustering quality but a surrogate for external agreement in settings where benchmark labels define the reference partition offline. This is a supervised surrogate-learning setup for unsupervised deployment (not circular, but bounded): the method inherits the notion of quality used in training. Multi-target analysis (Table 18, Appendix) shows robustness across AMI, ARI, NMI, and V-measure, but we do not claim robustness to arbitrary task-specific criteria. Pointwise regression and ranking losses achieve similar regret (Table 17, Appendix), suggesting the formulation matters more than the loss.

**Label semantics.**    We do not claim that benchmark class labels define the uniquely correct clustering of a dataset. METAIVM learns to predict agreement with the reference partitions used in training. This is appropriate when users want clusters aligned with semantic labels similar to those in the training benchmark (e.g., disease subtypes, object classes, curated taxonomies). It is less appropriate when the desired clustering is task-specific, multi-resolution, or orthogonal to class labels. In such cases, the relevant notion of quality must be encoded in the training distribution.

**Benchmark dependence and generalization.**    Our benchmark is synthetic-heavy (123/223 datasets) by design for controlled geometry coverage, not as a natural-frequency sample. The transfer asymmetry (synthetic→real: 0.100, real→synthetic: 0.238) confirms distribution dependence. On real data alone, METAIVM achieves 0.070 regret vs. CH's 0.207 on 91 OpenML datasets; on 8 hard real datasets (CH regret > 0.3), METAIVM wins all 8. The graph extension provides evidence beyond Euclidean data. Limited non-tabular evidence (5 text, 4 image) tempers cross-modality claims. Deployment cost is negligible: $< 1\,\mathrm{s}$ feature extraction, $< 0.2\,\mathrm{ms}$ inference, cheaper than Silhouette (Table 22, Appendix).

**Limitations.**    (1) Requires offline training on labeled datasets (mitigated: can use purely synthetic data). (2) Predicts external agreement (AMI), one operationalization of quality; domain-specific criteria may differ. (3) Limited evidence on specialized domains (genomics, time series). (4) Main benchmark omits Spectral Clustering ($O(n^3)$); post-hoc inclusion increases regret by only 0.005. (5) Struggles on high-noise, subspace, and extreme-imbalance datasets (19% of test cases above regret 0.15).

## 8  Conclusion

On our benchmark, external-agreement-based clustering model selection is substantially better served by a learned surrogate than by any fixed IVM. METAIVM reduces selection regret by 67% (Wilcoxon $p < 10^{-12}$), with improvements that are robust across metrics and models and extend to graph community detection. Principled controls confirm that the gains come from the per-partition formulation and feature space, not from learning alone or the inclusion of IVMs. Performance depends on the training distribution, and generalization to domains far from our benchmark remains an open question. We will release all data, code, and pre-trained models.

**Ethics and reproducibility.**    We foresee no significant negative societal impact. The benchmark uses only publicly available datasets. All code, datasets, pre-trained models, and experiment scripts will be released as open-source. The benchmark is fully reproducible with fixed random seeds (0–4) for all splits.

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

## A  Complete Feature Descriptions

Table 7 lists all 56 features extracted for each (dataset, partition) pair, grouped by source.

## B  Feature Importance

Table 8 shows the complete XGBoost feature importance ranking. The top 5 features account for 62% of total importance; the top 15 account for 84%.

Table 7: Complete feature list. All features are computed without access to ground-truth labels.

| Feature | Description |
| --- | --- |
| *Partition statistics (12 features)* | |
| n_clusters | Number of clusters in the partition |
| noise_fraction | Fraction of points assigned to noise ($-1$ label) |
| cluster_size_{min,max,mean,std,median} | Statistics of cluster sizes |
| size_ratio | Ratio of largest to smallest cluster |
| entropy | Shannon entropy of the cluster size distribution |
| gini | Gini impurity of cluster assignments |
| imbalance | Deviation from uniform cluster sizes |
| singleton_fraction | Fraction of clusters with a single point |
| *Dataset descriptors (14 features)* | |
| ds_n_samples, ds_n_features | Dataset dimensions |
| ds_log_n, ds_log_d | Log-transformed dimensions |
| ds_d_over_n | Dimensionality ratio $d/n$ |
| ds_mean_of_{means,stds,skewness,kurtosis} | Feature-wise distribution statistics (averaged) |
| ds_sparsity | Fraction of near-zero entries |
| ds_intrinsic_dim_pca95 | PCA components for 95% variance |
| ds_pairwise_dist_{mean,std,median} | Pairwise distance statistics (subsampled to 2k points) |
| *Partition-data interaction features (10 features)* | |
| within_cluster_var_{mean,std} | Within-cluster variance statistics |
| between_cluster_var | Between-cluster variance |
| dispersion_ratio | Between / within variance ratio |
| cluster_density_{mean,std,max} | Average pairwise distance within clusters |
| centroid_sep_{min,mean,std} | Pairwise centroid distance statistics |
| *Graph-based features (6 features)* | |
| graph_cut_fraction | Fraction of $k$NN edges crossing cluster boundaries |
| graph_normalized_cut_proxy | Proxy for the normalized graph cut |
| graph_within_cluster_edge_density | Mean edge density within clusters |
| graph_within_edge_density_std | Std of within-cluster edge density |
| graph_conductance_{min,mean} | Graph conductance statistics |
| *IVM features (3 features)* | |
| ivm_silhouette | Silhouette Coefficient |
| ivm_calinski_harabasz | Calinski-Harabasz Index |
| ivm_davies_bouldin_neg | Negated Davies-Bouldin Index |
| *Algorithm encoding (11 features)* | |
| algo_{KMeans,...,Agglom.} | One-hot encoding of algorithm family (6) |
| hp_{n_clusters,...,min_cluster_size} | Numeric hyperparameter values (5) |

## C  Clustering Algorithm Configurations

Table 9 details the complete hyperparameter grid for each clustering algorithm (91 configurations in total). At runtime, SpectralClustering was skipped due to $O(n^3)$ complexity, and AgglomerativeClustering was skipped for datasets with $n > 10,000$ due to $O(n^2)$ memory. After these filters, the effective count ranges from 19 to 79 per dataset (mean: 75.7; lower counts occur for the largest datasets where additional GMM configurations also fail to converge), yielding 16,889 total runs across 223 datasets.

## D  Per-Seed Results

Table 10 shows selection regret and Spearman $\rho$ for each method on each of the 5 random seeds. Results are consistent across seeds, with low variance for METAIVM.

Table 8: Complete feature importance ranking (XGBoost gain). Features below 0.5% omitted for brevity.

| Rank | Feature | Importance |
|---:|---|---:|
| 1 | ivm_calinski_harabasz | 34.96% |
| 2 | ds_mean_of_kurtosis | 9.75% |
| 3 | graph_conductance_min | 7.96% |
| 4 | cluster_size_mean | 4.64% |
| 5 | ds_sparsity | 4.43% |
| 6 | entropy | 4.31% |
| 7 | ds_pairwise_dist_std | 4.08% |
| 8 | ds_mean_of_skewness | 2.57% |
| 9 | ds_d_over_n | 2.15% |
| 10 | between_cluster_var | 1.81% |
| 11 | noise_fraction | 1.72% |
| 12 | ds_n_features | 1.62% |
| 13 | gini | 1.61% |
| 14 | ivm_silhouette | 1.59% |
| 15 | ds_n_samples | 1.48% |
| 16 | singleton_fraction | 1.43% |
| 17 | cluster_density_mean | 1.18% |
| 18 | ds_pairwise_dist_median | 1.08% |
| 19 | ds_mean_of_means | 0.92% |
| 20 | algo_GaussianMixture | 0.82% |

Table 9: Clustering algorithm hyperparameter grids. Total configurations: 91 per dataset (before filtering).

| Algorithm | Hyperparameters | Configs |
|---|---|---:|
| KMeans | $k \in \{2, 3, 5, 7, 10, 15, 20, 30\}$, n_init $= 10$ | 8 |
| Gaussian Mixture | $k \in \{2, 3, 5, 7, 10, 15, 20\}$, cov $\in \{\text{full}, \text{diag}\}$ | 14 |
| Spectral | $k \in \{2, 3, 5, 7, 10, 15\}$, affinity $\in \{\text{rbf}, \text{nn}\}$ | 12* |
| DBSCAN | $\varepsilon \in \{0.1, 0.3, 0.5, 1.0, 2.0, 5.0\}$, min_samp $\in \{3, 5, 10, 20\}$ | 24 |
| HDBSCAN | min_cluster $\in \{5, 10, 20, 50\}$, min_samp $\in \{1, 5, 10\}$ | 12 |
| Agglomerative | $k \in \{2, 3, 5, 7, 10, 15, 20\}$, link $\in \{\text{ward}, \text{comp.}, \text{avg.}\}$ | 21[†] |

*Skipped at runtime (cubic complexity). [†]Skipped for $n > 10{,}000$ (quadratic memory).

# E  Dataset Collection Details

Our benchmark comprises 223 datasets from four sources. Table 11 provides summary statistics.

**OpenML datasets (91).**   We selected datasets from the OpenML CC18 benchmark suite (Bischl et al., 2016) plus 32 additional hand-selected datasets chosen for domain diversity (biology, finance, physics, social sciences, computer vision). Datasets with $n > 20{,}000$ samples were subsampled uniformly at random; those with $d > 200$ features were reduced to 50 dimensions via PCA. Three datasets (`openml_3`, `openml_46`, `openml_50`) were excluded due to all-NaN features after preprocessing.

**Synthetic datasets (123).**   Generated using scikit-learn generators and custom code to systematically vary:

- **Dimensionality**: $d \in \{2, 3, 5, 10, 15, 20, 30, 50, 100, 200\}$

- **Number of clusters**: $k \in \{2, 3, 4, 5, 6, 8, 10, 12, 15, 20\}$

- **Cluster overlap**: Low ($\sigma = 0.5$), medium ($\sigma = 1.5$), high ($\sigma = 3.0$)

- **Cluster shape**: Spherical (blobs), ellipsoidal (random covariance), manifold (Swiss roll, S-curve), non-convex (moons, circles)

Table 10: Per-seed results for all methods (selection regret / Spearman $\rho$).

| Method | Seed 0 | Seed 1 | Seed 2 | Seed 3 | Seed 4 |
|---|---|---|---|---|---|
| Random | .170 / .035 | .160 / .030 | .212 / .006 | .171 / .007 | .174 / .019 |
| Global best | .236 / .106 | .159 / .061 | .352 / .024 | .327 / .010 | .130 / .063 |
| Silhouette | .292 / .375 | .390 / .369 | .390 / .478 | .361 / .467 | .276 / .476 |
| Calinski-Harabasz | .195 / .589 | .249 / .573 | .270 / .634 | .230 / .604 | .142 / .621 |
| Davies-Bouldin | .297 / .337 | .393 / .329 | .395 / .427 | .365 / .455 | .277 / .448 |
| CH Adjusted | .229 / .513 | .288 / .490 | .282 / .555 | .282 / .535 | .207 / .559 |
| Ridge | .093 / .617 | .111 / .620 | .109 / .710 | .072 / .693 | .055 / .669 |
| MLP | .070 / .616 | .083 / .647 | .074 / .761 | .066 / .716 | .068 / .685 |
| **MetaIVM (XGB)** | **.064 / .675** | **.072 / .752** | **.078 / .801** | **.074 / .745** | **.065 / .738** |

Table 11: Dataset collection summary statistics (across 223 datasets with clustering runs).

| Statistic | Min | Median | Mean | Max |
|---|---|---|---|---|
| Samples ($n$) | 150 | 1,200 | 4,438 | 48,842 |
| Features ($d$) | 2 | 14 | 33 | 500 |
| Classes ($k$) | 2 | 4 | 6.3 | 50 |
| Runs per dataset | 19 | 79 | 75.7 | 91 |

- **Imbalance**: Mild (1:1.5:2.5), moderate (1:3:16), extreme (1:1.5:2.5:45), many-small (6 clusters with one dominant)

- **Noise**: 5%, 15%, 25% uniform background noise

- **Subspace structure**: Relevant features in $\{2, 3, 5\}$ out of $\{50, 100, 200\}$ total dimensions

**Text datasets (5).** Five subsets of the 20Newsgroups corpus: all categories, science, politics+religion, computers, and recreation. Text was converted to TF-IDF vectors (5,000-word vocabulary, English stop words removed) and reduced to 50 dimensions via truncated SVD. Subsampled to 2,000–5,000 documents.

**Image datasets (4).** MNIST and Fashion-MNIST digit/clothing images, each at 2,000 and 5,000 samples. Pixel features (784 dimensions) were reduced to 50 dimensions via PCA.

## F  Preprocessing Pipeline

Each dataset undergoes the following preprocessing before clustering:

1. **Missing value handling**: Columns with >50% missing values are dropped. Remaining missing values are imputed with column medians.

2. **Constant feature removal**: Features with zero variance are dropped.

3. **Standardization**: All features are standardized to zero mean, unit variance.

4. **Graph construction**: $k$-nearest-neighbor graphs are computed for $k \in \{10, 15, 20\}$ using Euclidean distance, stored as sparse adjacency matrices.

## G  Zero Feature Engineering Details

Table 12 compares approaches with varying levels of feature engineering. All results are preliminary (single architectures, no hyperparameter search).

Table 12: Impact of feature engineering. Even zero-engineering approaches (graph embeddings, GNN) substantially outperform all classical IVMs.

| Method | Engineering | Regret ↓ | Spearman $\rho$ ↑ |
|---|---|---|---|
| Best IVM (CH) | n/a | $.217 \pm .237$ | $.604 \pm .320$ |
| XGBoost (graph spectral emb.) | none | $.105 \pm .130$ | $.667 \pm .290$ |
| GNN (end-to-end) | none | $.131 \pm .193$ | $.705 \pm .229$ |
| XGBoost (auto-features, 18) | minimal | $.083 \pm .120$ | $.743 \pm .270$ |
| **MetaIVM (hand-crafted, 56)** | **heavy** | $\mathbf{.073 \pm .103}$ | $\mathbf{.758 \pm .302}$ |

The GNN operates directly on the $k$NN graph with partition labels as 32-dimensional one-hot node features (3-layer GIN, 128 hidden dims; see Appendix H for details). Despite no feature engineering, it achieves $\rho = 0.705$, above the best IVM ($\rho = 0.604$). Graph spectral embeddings (top-16 eigenvectors of the normalized graph Laplacian, concatenated with partition statistics) fed to XGBoost achieve lower regret (0.105) but lower $\rho$ (0.667). The 18 auto-features (cluster sizes, within/between variance, centroid distances) close most of the gap to the full model. Hand-crafted features provide a further 12% improvement, primarily through graph conductance and dataset-level descriptors (kurtosis, sparsity).

## H  Model Training Details

**XGBoost configuration.**  We use the `xgboost` Python package with the following hyperparameter search: max depth $\in \{3, 5, 7\}$, learning rate $\in \{0.01, 0.05, 0.1\}$ (9 combinations total). Each configuration is trained with up to 500 estimators and early stopping (patience 20 rounds) on the validation set, using RMSE as the stopping criterion. The best configuration (lowest validation MSE) is then refitted on the combined train+val data using the optimal number of estimators from early stopping. Missing feature values (e.g., IVM scores for degenerate partitions) are handled natively by XGBoost's sparse-aware split finding.

**MLP configuration.**  Two hidden layers (256, 128 units), ReLU activations, Adam optimizer ($\mathrm{lr} = 10^{-3}$), max 500 epochs with early stopping (patience 20). The validation set is used for early stopping via scikit-learn's built-in validation fraction mechanism. Features are standardized to zero mean and unit variance. Missing values are imputed with zeros before scaling.

**Ridge regression.**  $L_2$-regularized linear regression with regularization strength $\alpha$ selected from $\{0.01, 0.1, 1.0, 10.0, 100.0\}$ via validation $R^2$ score. The best $\alpha$ is used to refit on the combined train+val data. Features are standardized as above.

**GNN configuration.**  3 GIN layers with 128 hidden dimensions, residual connections, batch normalization, 10% dropout. Partition labels encoded as 32-dimensional one-hot vectors (with hash collision for $k > 31$). Node features truncated/padded to 50 dimensions. Partition-aware pooling: mean within each cluster, then mean across cluster representations, concatenated with global mean pool. Adam optimizer ($\mathrm{lr} = 10^{-3}$), batch size 32, max 80 epochs, early stopping patience 15.

## I  Evaluation Protocol

**Dataset-level splits.**  For each random seed $s \in \{0, 1, 2, 3, 4\}$, we randomly partition the 223 datasets into 60% train (134 datasets), 20% validation (44 datasets), and 20% test (45 datasets). All clustering runs for a given dataset are assigned to the same split, preventing any information leakage between train and test.

**Metric definitions.**

- **Selection regret**: $\mathrm{Regret}(d) = \max_j \mathrm{AMI}_j - \mathrm{AMI}_{j^*}$ where $j^* = \arg\max_j \hat{q}_j$. Averaged across test datasets, then across seeds.

- **$\varepsilon$-success**: Fraction of test datasets satisfying Regret$(d) \leq 0.01$, i.e., where the selected partition is within an absolute AMI gap of 0.01 from the best available partition.

- **NDCG@5**: Normalized discounted cumulative gain using true AMI as relevance, evaluated on the top 5 predicted partitions.

- **Spearman $\rho$**: Rank correlation between predicted scores and true AMI, computed per dataset, then averaged.

- **Kendall $\tau$**: Concordance probability for all pairwise comparisons within a dataset, then averaged.

**IVM baselines.** Classical IVMs require no training. For each test dataset, the partition with the highest IVM score (or lowest, for Davies-Bouldin) is selected. The adjusted CH index (Jeon et al., 2025) is computed using the official `btwim` Python library with 20 Monte Carlo iterations for the permutation-based normalization, applied pairwise across all cluster pairs. We follow the default setting of 20 iterations from Jeon et al.; spot checks with 50 iterations showed <0.5% change in mean CH$_A$ values, indicating adequate convergence for model selection purposes.

## J  Formal Illustration: Why Fixed IVMs Disagree

We provide a constructive example showing that compactness-separation IVMs cannot be simultaneously rank-consistent across heterogeneous dataset families. This is not a general impossibility theorem (it applies specifically to the class of IVMs defined below) but it formalizes the intuition motivating a learned evaluator. Figure 2 illustrates the construction visually.

[Compactness-Separation IVM] A *compactness-separation IVM* is a function $\phi : (\mathcal{X}, \pi) \mapsto \mathbb{R}$ that depends on the data $\mathcal{X} = \{x_1, \ldots, x_n\} \subset \mathbb{R}^d$ and partition $\pi : [n] \to [k]$ only through the within-cluster dispersion

$$W(\mathcal{X}, \pi) = \frac{1}{n} \sum_{j=1}^{k} \sum_{i:\pi(i)=j} \|x_i - \mu_j\|^2, \quad \mu_j = \frac{1}{|C_j|} \sum_{i \in C_j} x_i$$

and the between-cluster separation

$$B(\mathcal{X}, \pi) = \frac{1}{n} \sum_{j=1}^{k} |C_j| \cdot \|\mu_j - \bar{x}\|^2, \quad \bar{x} = \frac{1}{n} \sum_i x_i,$$

such that $\phi$ is non-decreasing in $B$ and non-increasing in $W$ (or equivalently, non-decreasing in $B/W$).

This class includes the Calinski-Harabasz index (CH $= \frac{B/(k-1)}{W/(n-k)}$), the Davies-Bouldin index (inversely related to $B/W$ per cluster pair), and any monotone function of the variance ratio. The Silhouette Coefficient, while not strictly of this form, is closely related: it rewards points that are much closer to their own cluster centroid than to the nearest other centroid, which correlates with low $W$ and high $B$.

[Rank-Consistency] An IVM $\phi$ is *rank-consistent with respect to an external quality measure $q$* on a dataset family $\mathcal{F}$ if, for all datasets $\mathcal{X} \in \mathcal{F}$ and all partition pairs $\pi_a, \pi_b$:

$$q(\pi_a, y^*) > q(\pi_b, y^*) \implies \phi(\mathcal{X}, \pi_a) > \phi(\mathcal{X}, \pi_b),$$

where $y^*$ denotes the ground-truth labels and $q$ is an external measure such as AMI.

[Impossibility of Universal Rank-Consistency] No compactness-separation IVM $\phi$ (Definition J) is rank-consistent (Definition J) simultaneously on the family $\mathcal{F}_{\text{sph}}$ of isotropic Gaussian mixtures and the family $\mathcal{F}_{\text{ell}}$ of mixtures with heterogeneous covariance structure.

We construct explicit datasets and partitions in each family.

**Family $\mathcal{F}_{\text{sph}}$.** Let $\mathcal{X}_1 \subset \mathbb{R}^2$ consist of $n = 300$ points drawn from three isotropic Gaussians with $\sigma = 0.3$ and centers $c_1 = (0,0)$, $c_2 = (10,0)$, $c_3 = (5, 8.66)$ (equilateral triangle, side length 10). Consider:

**Proposition 1: No compactness–separation IVM is universally rank-consistent**

*CH (and any monotone function of B/W) ranks correctly on spherical clusters but is fooled by elongated ones.*

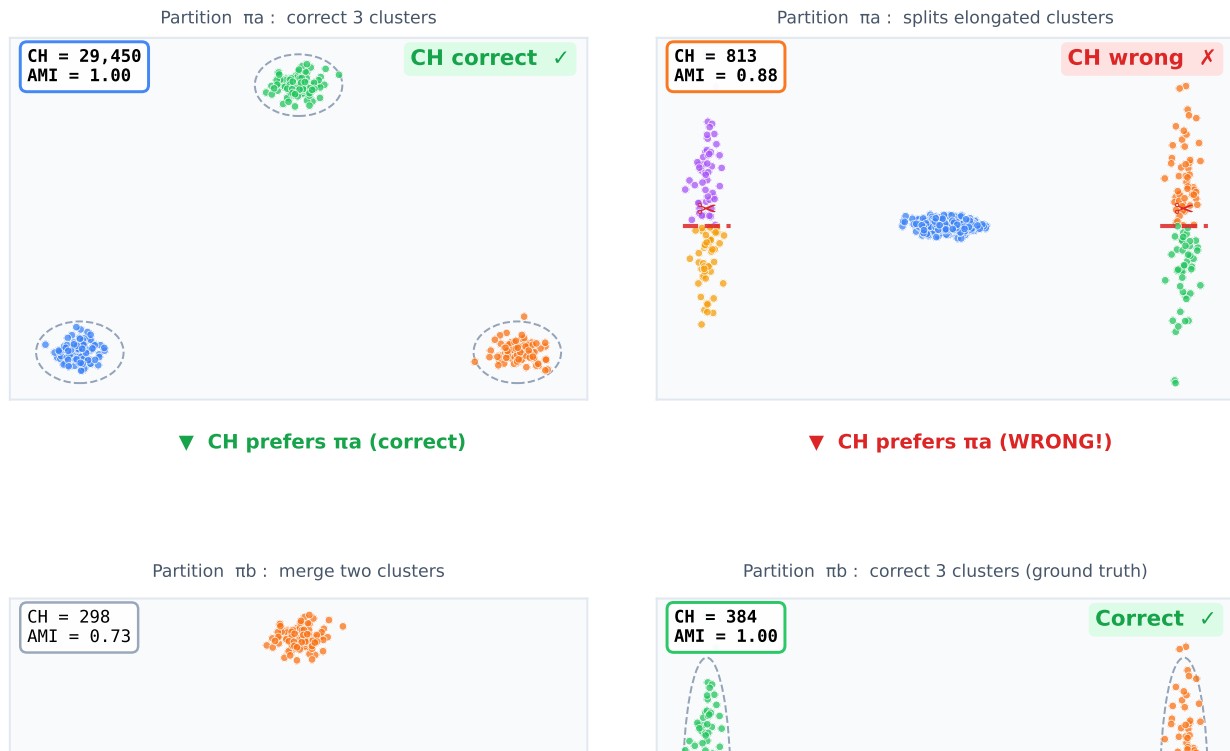

Figure 2: Visual illustration of Proposition J. **Left:** On isotropic Gaussians ($\mathcal{F}_{\text{sph}}$), CH correctly ranks the 3-cluster partition (CH=29,450) above the merged partition (CH=298). **Right:** On heterogeneous covariance ($\mathcal{F}_{\text{ell}}$), CH incorrectly prefers the partition that splits elongated clusters (CH=813, dashed red lines) over the ground truth (CH=384, AMI=1.0), because splitting reduces within-cluster variance.

- $\pi_a$: the correct 3-cluster assignment (each point to its generating center);

- $\pi_b$: a 2-cluster assignment that merges clusters 2 and 3.

Since the clusters are well-separated and isotropic, $\pi_a$ has lower within-cluster dispersion $W$ and higher between-cluster separation $B$ than $\pi_b$. Thus for any CS-IVM $\phi$: $\phi(\mathcal{X}_1, \pi_a) > \phi(\mathcal{X}_1, \pi_b)$. This agrees with external quality: $\text{AMI}(\pi_a, y^*) > \text{AMI}(\pi_b, y^*)$ since $\pi_a$ is the ground truth.

**Family $\mathcal{F}_{\text{ell}}$.** Let $\mathcal{X}_2 \subset \mathbb{R}^2$ consist of three groups:

- Cluster 1: $n_1 = 400$ points from $\mathcal{N}((0,0), 0.2^2 I)$, a dense, compact sphere;

- Cluster 2: $n_2 = 100$ points from $\mathcal{N}((3,0), \text{diag}(0.1^2, 2.5^2))$, a vertically elongated ellipsoid;

- Cluster 3: $n_3 = 100$ points from $\mathcal{N}((-3,0), \text{diag}(0.1^2, 2.5^2))$, another vertically elongated ellipsoid.

Consider:

- $\pi_a$: a partition that assigns the dense sphere correctly but splits each ellipsoid horizontally at $y = 0$, assigning points with $y > 0$ to one cluster and $y < 0$ to another (4 clusters total). This reduces within-cluster variance $W$ (each half-ellipsoid is more compact) and increases $B$ (the half-ellipsoid centroids are more spread).

- $\pi_b$: the correct 3-cluster assignment matching the generating distribution.

For $\pi_a$: $W$ is low because splitting elongated clusters reduces variance; $B$ is high because the split produces centroids that are further apart vertically. For $\pi_b$: $W$ is higher because each elongated cluster has large variance along its major axis. Therefore $\phi(\mathcal{X}_2, \pi_a) > \phi(\mathcal{X}_2, \pi_b)$ for any CS-IVM.

However, $\pi_b$ is the ground truth, so $\mathrm{AMI}(\pi_b, y^*) = 1.0 > \mathrm{AMI}(\pi_a, y^*)$.

**Combining.** The CS-IVM $\phi$ ranks correctly on $\mathcal{F}_{\mathrm{sph}}$ ($\phi$ prefers $\pi_a$, which has higher AMI) but incorrectly on $\mathcal{F}_{\mathrm{ell}}$ ($\phi$ prefers $\pi_a$, which has *lower* AMI). Since $\phi$ is monotone in $B/W$ by definition, no choice of $\phi$ within this class resolves the disagreement.

**Remark.** The construction exploits a fundamental limitation: CS-IVMs reward reducing within-cluster variance, but splitting an elongated cluster always reduces $W$ regardless of whether the split is meaningful. Any IVM that is monotone in the variance ratio is susceptible to this failure mode. A learned evaluator can distinguish these cases by conditioning on dataset properties; for instance, graph conductance on the $k$NN graph detects whether a split respects the data manifold structure (low conductance) or cuts through it (high conductance).

**Empirical verification.** We verify this prediction in our benchmark (averaged across 5 seeds): CH achieves positive Spearman $\rho$ on 94% of test datasets but *negative $\rho$* on 6%, meaning it actively prefers worse partitions on those datasets. METAIVM achieves positive $\rho$ on 98% and negative on only 2%. While CH is correct directionally on most datasets, the proposition shows it is not *rank-consistent*: it may prefer the wrong partition even when positively correlated overall. The 6% failure rate for CH, though modest, corresponds to real datasets with non-spherical structure and imbalanced cluster sizes, consistent with the mechanism identified in Proposition J.

## K   Compact Feature Set: Selection Protocol

The 15-feature compact model is constructed as follows:

1. For each of the 5 training seeds, train XGBoost on the training fold and extract feature importances (gain-based).

2. Rank features by importance for each seed.

3. Select the 15 features that appear in the top-15 for at least 3 out of 5 seeds.

The resulting 15 features are: `ivm_calinski_harabasz`, `ds_mean_of_kurtosis`, `graph_conductance_min`, `entropy`, `ds_mean_of_skewness`, `ds_n_samples`, `ds_pairwise_dist_std`, `cluster_size_mean`, `graph_conductance_mean`, `ds_pairwise_dist_mean`, `between_cluster_var`, `ds_pairwise_dist_median`, `ivm_silhouette`, `ds_d_over_n`, `noise_fraction`.

Eight of these features are stable across *all* 5 seeds: `ivm_calinski_harabasz`, `ds_mean_of_kurtosis`, `graph_conductance_min`, `entropy`, `ds_mean_of_skewness`, `ds_n_samples`, `ds_pairwise_dist_std`, `cluster_size_mean`. Importantly, feature selection is performed exclusively on training folds; test data is never used for selection.

## L   Additional Experimental Tables

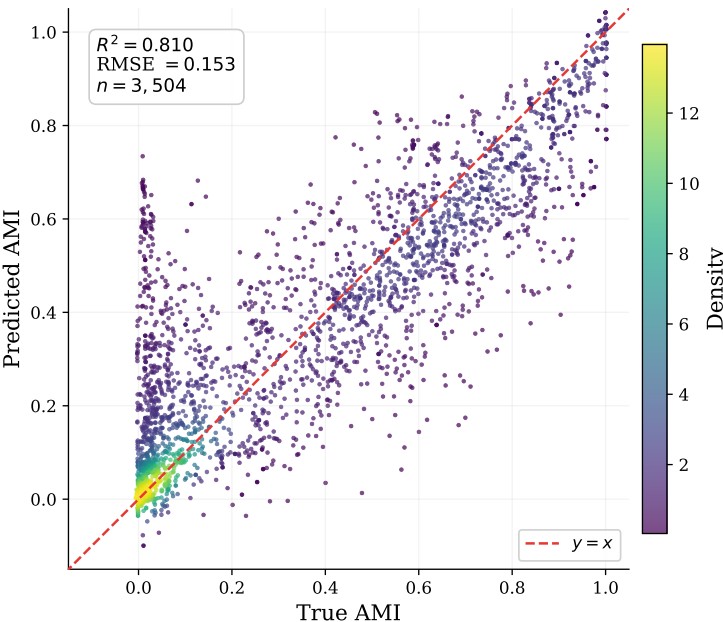

Figure 3: Predicted vs. true AMI for all test (dataset, partition) pairs across 5 seeds ($R^2 = 0.810$, RMSE $= 0.153$, $n = 3,504$). Points near the diagonal indicate accurate prediction. The model captures the full range of external agreement, from poor (AMI $\approx 0$) to excellent (AMI $\approx 1$).

Table 13: Cross-domain transfer. METAIVM trained on one domain, tested on another. Best IVM baseline shown for comparison.

| Scenario | MetaIVM | | Best IVM (CH) | |
|---|---|---|---|---|
| | Regret ↓ | $\rho$ ↑ | Regret ↓ | $\rho$ ↑ |
| In-distribution (60/20/20) | .071 | .742 | .217 | .604 |
| Synthetic → real | .100 | .678 | .235 | .489 |
| Synthetic → OpenML | .100 | .657 | .218 | .466 |
| Synthetic → image+text | .105 | .887 | .398 | .722 |
| Real → synthetic | .238 | .607 | .193 | .761 |
| OpenML → synthetic | .209 | .590 | .193 | .761 |

Table 14: Leave-one-algorithm-out: impact of removing each algorithm family from training.

| Held-out Algorithm | Regret (without) | Δ Regret |
|---|---|---|
| KMeans | $.047 \pm .084$ | $+.004$ |
| Gaussian Mixture | $.047 \pm .075$ | $+.004$ |
| DBSCAN | $.043 \pm .107$ | $+.023$ |
| HDBSCAN | $.022 \pm .044$ | $-.001$ |
| Agglomerative | $.049 \pm .065$ | $+.004$ |

Table 15: Per-dataset Spearman correlation between predicted scores and true AMI (averaged across 223 test datasets, 5 seeds).

| Method | Spearman $\rho$ |
|---|---|
| METAIVM (predicted) | $\mathbf{0.755 \pm 0.253}$ |
| CH Adjusted (Jeon et al., 2025) | $0.269 \pm 0.435$ |
| Calinski-Harabasz | $0.395 \pm 0.416$ |
| Silhouette | $0.062 \pm 0.552$ |
| Davies-Bouldin (neg) | $0.011 \pm 0.528$ |

Table 16: Model selection on 7 real-world attributed graphs (189 runs, leave-one-out). Classical IVMs use only node attributes; Silhouette and CH are anti-correlated with quality.

| Method | Regret ↓ | Spearman $\rho$ ↑ |
|---|---|---|
| *Attribute-only (ignore graph)* | | |
| Silhouette (attributes) | .395 | $-.588$ |
| Calinski-Harabasz (attributes) | .350 | $-.481$ |
| Davies-Bouldin (attributes) | .039 | .548 |
| *Graph-only (ignore attributes)* | | |
| Modularity | .028 | .592 |
| *Learned (leave-one-dataset-out)* | | |
| METAIVM (graph only, 6 train) | .039 | .491 |
| METAIVM (graph + attr, 6 train) | .029 | .513 |
| **MetaIVM (graph only, LOO)** | **.004** | **.657** |
| *Transfer from synthetic graphs* | | |
| METAIVM (SBM→Attr, 80 train) | .054 | .288 |

Table 17: Training objective comparison under the paper's exact 5-seed 60/20/20 protocol. All three objectives use the same XGBoost architecture and hyperparameters. Differences are modest relative to the gap over classical IVMs (CH regret $\geq 0.20$).

| Training Objective | Regret ↓ | Spearman $\rho$ ↑ |
|---|---|---|
| Pointwise regression (MSE) | $.068 \pm .012$ | $.761 \pm .033$ |
| LambdaMART (rank:ndcg) | $.062 \pm .017$ | $.782 \pm .021$ |
| Pairwise (rank:pairwise) | $.065 \pm .015$ | $.796 \pm .027$ |
| *Best IVM (CH)* | *.217 ± .237* | *.604 ± .320* |

Table 18: Selection regret across four external quality metrics (5 seeds, 28-feature set). The AMI regret here (0.076) differs slightly from Table 1 (0.071) because each metric uses a separately trained model and the averaging scope differs.

| Method | AMI | ARI | NMI | V-measure |
|---|---|---|---|---|
| METAIVM (XGBoost) | **.076** | **.092** | **.072** | **.072** |
| Ridge | .123 | .152 | .125 | .125 |
| Calinski-Harabasz | .217 | .250 | .219 | .219 |
| Silhouette | .342 | .380 | .340 | .340 |
| Davies-Bouldin | .251 | .324 | .254 | .254 |

Table 19: Paired Wilcoxon signed-rank tests: MetaIVM (XGBoost) vs. each baseline. Per-dataset regret averaged across 5 seeds, 158 test datasets.

| Comparison | XGB wins | Baseline wins | Ties | $p$-value |
|---|---|---|---|---|
| vs. Calinski-Harabasz | 110 | 39 | 9 | $4.1 \times 10^{-13}$ |
| vs. Silhouette | 139 | 13 | 6 | $1.5 \times 10^{-24}$ |
| vs. Davies-Bouldin | 137 | 15 | 6 | $2.4 \times 10^{-24}$ |
| vs. Ridge | 74 | 69 | 15 | $2.2 \times 10^{-2}$ |

Table 20: Learning curves: regret and Spearman $\rho$ vs. training set size (fraction of 133 datasets, 5 seeds).

| Frac. | $N_{\text{train}}$ | XGBoost | | Ridge | |
|---|---|---|---|---|---|
| | | Regret $\downarrow$ | $\rho \uparrow$ | Regret $\downarrow$ | $\rho \uparrow$ |
| 10% | 13 | .116 | .717 | .108 | .622 |
| 20% | 26 | .117 | .696 | .108 | .624 |
| 30% | 39 | .085 | .739 | .107 | .625 |
| 50% | 66 | .083 | .758 | .110 | .629 |
| 70% | 93 | .085 | .759 | .126 | .625 |
| 100% | 133 | .076 | .760 | .123 | .634 |

Table 21: Top failure cases: datasets with highest average selection regret for MetaIVM (XGBoost), across 5 seeds.

| Dataset | Regret | Best AMI | Failure mode |
|---|---|---|---|
| synth_uniform_noise_2d | 0.658 | 0.989 | High noise, ambiguous structure |
| synth_subspace_5of50 | 0.575 | 0.996 | Subspace clustering |
| synth_imb_many_small_d15 | 0.429 | 1.000 | Extreme imbalance |
| synth_blobs_many_2d | 0.379 | 0.992 | Many clusters, 2D |
| openml_1040 (sylva_prior) | 0.367 | 0.478 | High-d (108), low signal |
| synth_moons_noise0.15 | 0.327 | 0.818 | Non-convex + noise |
| synth_mixed_shapes_2d | 0.309 | 1.000 | Mixed cluster shapes |

Table 22: Asymptotic complexity per dataset with $n$ samples, $d$ features, $k$ clusters, and $M$ candidate partitions. MetaIVM is cheaper than Silhouette and comparable to CH/DB.

| Component | Complexity | Note |
|---|---|---|
| *MetaIVM (deployment)* | | |
| $k$NN graph | $O(n \log n \cdot d)$ | One-time per dataset, reusable |
| Feature extraction | $O(M \cdot n \cdot k)$ | Per partition: cluster stats + graph features |
| Model inference | $O(M \cdot T \cdot d_f)$ | $T$=trees, $d_f$=28 features; $< 1\,\text{ms}$ |
| **Total** | $O(n \log n \cdot d + M \cdot n \cdot k)$ | |
| *Classical IVMs* | | |
| Silhouette | $O(M \cdot n^2)$ | Pairwise distance matrix per partition |
| Calinski-Harabasz | $O(M \cdot n \cdot k)$ | Comparable to MetaIVM |
| Davies-Bouldin | $O(M \cdot n \cdot k)$ | Comparable to MetaIVM |

Table 23: Candidate-pool robustness. MᴇᴛᴀIVM trained on the full pool, tested on modified pools at test time. MᴇᴛᴀIVM outperforms CH across all 11 configurations, supporting the view that the learned surrogate is not specialized to a particular candidate menu.

| Pool configuration | MetaIVM regret ↓ | CH regret |
|---|---|---|
| Full pool (all algorithms) | .057 | .199 |
| Drop KMeans | .053 | .205 |
| Drop GaussianMixture | .048 | .183 |
| Drop DBSCAN | .055 | .134 |
| Drop HDBSCAN | .059 | .191 |
| Drop AgglomerativeClustering | .054 | .195 |
| KMeans only | .036 | .085 |
| Density-based only (DBSCAN+HDBSCAN) | .023 | .119 |
| Random 50% subsample | .055 | .174 |
| Only $k \geq 5$ | .054 | .139 |
| Only $k \leq 3$ | .042 | .127 |

