# OpenReview forum: "Meta-Learned Surrogates for Clustering Model Selection"
_TMLR — Under review for TMLR_

### Review · Reviewer_WMA6 · 2026-06-02

**Summary Of Contributions:**

This paper studies the evaluating of clustering model selection. The paper identifies the gap in knowledge that previous method rely on IVMs, which does not necessarily align with external agreements. To cope with this issue, the paper proposes MetaIVM, which learns a model to output the score of clustering result. In particular MetaIVM is a framework that can incorporate multiple models and is trained on a collection of different dataset and algorithm. The paper provides empirical evidence that MetaIVM performs better than baseline when used to select clustering model.

**Strengths**

1. The design to score combinations of dataset and partition is meaningful for selecting specialized algorithms for different dataset.
2. The paper provided ablation study that the learned model does not solely rely on IVMs.
3. The paper shows promising results.

**Weaknesses**

1. The proposed model is largely based on hand-crafted features. Although the paper provides evidence that the IVMs are not the sole contributor, a thorough statistical investigation of how important each feature is might be necessary.
2. It is not clear whether the learned MetaIVM model can generalize to unseen datasets and clustering algorithms.
3. The paper's description of the experimental setup in Section 5.1-5.4 are quite vague.
4. Although the collection of datasets is large in size, it consists of a lot of synthetic datasets which may not reflect real-world dataset. At least a report on the performance on synthetic and non-synthetic datasets is needed.

**Audience:**

Yes

**Audience Explanation:**

The need for a good scoring mechanism for clustering algorithms without true label is indeed important.

**Claims And Evidence:**

No

**Claims Explanation:**

I think the main issue is that the capability of the algorithm is not thoroughly tested across different scenarios. Please see weaknesses.

**Requested Changes:**

1. Added a study on the statistical significance of each feature is needed.
2. The paper needs to justify the performance on unseen dataset and algorithms.
3. The paper need to provide a better description of the experiment setting.
4. The paper need to report separate performance on synthetic and non-synthetic dataset.

---

### Review · Reviewer_dfk5 · 2026-06-05

**Summary Of Contributions:**

This paper reframes clustering model selection as a supervised problem. Instead of relying on fixed Internal Validity Measures, the authors propose MetaIVM, a model trained offline on labeled benchmarks that predicts the external agreement (primarily AMI) of an individual (dataset, partition) pair from observable features, then deployed without labels.

The central contributions are:

- A per-partition surrogate formulation for clustering model selection. Reframing model selection as predicting external agreement for individual (dataset, partition) pairs, rather than recommending an algorithm at the dataset level. This unifies algorithm choice, hyperparameter selection, and cluster-count selection in one framework, since any partition from any method can be scored.

- MetaIVM itself and strong empirical results. A meta-learned, model-agnostic surrogate (Ridge/MLP/XGBoost on the same features) trained offline on labeled benchmarks and deployed without labels, achieving a 67% reduction in selection regret over the best classical IVM on 223 datasets and 16,889 runs.

- Benchmark and evaluation. A benchmark of 223 datasets.

- Principled controls isolating the source of improvement. Showing that the gains come from the per-partition formulation and feature space rather than learning per se.

- Evidence of adaptive, context-dependent feature reliance. Demonstrating that the predictive signal shifts with geometry, which fixed IVMs cannot do.

- Robustness and transfer characterization.

- Extension to graph community detection. A setting where coordinate-based IVMs don't apply, where MetaIVM outperforms modularity-based selection.

## Strengths

**1. The core scientific claim is well-isolated.** The strongest feature of the paper is the control ladder in Section 6.2 (Table 2). Rather than only showing that MetaIVM beats IVMs, the authors separate *why*.

**2. The central result is robust and statistically sound.** The headline 67% regret reduction is averaged over 5 dataset-level splits with leakage-preventing splits.

**3. Practical relevance.** Negligible deployment cost (<1s feature extraction, <0.2ms inference), low sample complexity, model-agnosticism across Ridge/MLP/XGBoost, and a promised release of data, code, and pre-trained models make the work directly usable.

## Weaknesses

**1. Lack of theory.** There is no characterization of the conditions under which the *learned* surrogate succeeds. Given that, by a No-Free-Lunch argument, no selector can dominate across all problem distributions, the method's gains must rest on unstated structural assumptions about the distribution of clustering problems.

**2. Notational imprecision in the problem formulation.** The objective is written as learning $f(X_i, \pi_{ij})$, with $X_i$ defined as the dataset feature matrix, but in practice the model consumes an engineered 56-dimensional feature vector with heterogeneous dependencies (some features depend on the partition alone, some on the data alone, some on both). The feature-extraction step is not made explicit in the notation, and the feature-to-argument mapping must be reconstructed by the reader. This is a presentation issue, not a correctness one, but it obscures the paper's own central point that observing the partition is what drives the gains.

**3. Minor numerical-hygiene friction.** The headline regret figure wanders across tables (0.065 / 0.070 / 0.071 / 0.073 / 0.076) depending on feature set and averaging scope, and the test-set size shifts (223 / 158 / 46 / 45) across tables. The captions explain these, so it is not deceptive, but it requires the reader to track which configuration is being quoted at each point and is mildly distracting.

**Additional Comments:**

NA

**Audience:**

Yes

**Audience Explanation:**

Practitioners doing unsupervised clustering would be interested in this paper. Anyone selecting among algorithms/hyperparameters without labels currently reaches for Silhouette or CH by default. The finding that these correlate poorly with external agreement and fail catastrophically on a non-trivial fraction of datasets is directly actionable.

**Claims And Evidence:**

Yes

**Claims Explanation:**

The core scientific claim, i.e. that a per-partition learned surrogate substantially beats fixed IVMs, and that this is due to the formulation rather than learning or IVM-inclusion,  is supported by accurate, convincing, and well-controlled evidence. The evidence is convincing for what the paper actually claims; it would be overclaiming only if read as "universal clustering quality," which the Discussion explicitly disavows.

**Requested Changes:**

## *Notation*


We found the notation around the learned function slightly imprecise, and clarifying it would improve readability and, we think, reinforce the paper's main contribution.

In Section 3.1 the objective is written as learning $f(X_i, \pi_{ij}) \rightarrow \hat{q_{ij}}$, with $X_i \in \mathbb{R}^{n_i \times d_i}$ the dataset feature matrix and $\pi_{ij}$ a candidate partition. As written, this suggests $f$ maps the raw matrix and the raw assignment directly to a predicted AMI. In practice (Section 4.1, Table 7), the model instead consumes a fixed 56-dimensional feature vector computed from $(X_i, \pi_{ij})$. We would suggest making the feature-extraction step explicit in the notation — for example, define a feature map $\phi(X_i, \pi_{ij}) \in \mathbb{R}^{56}$ and write the predictor as $f(\phi(X_i, \pi_{ij}))$ — so that the engineered representation is clearly distinguished from the raw inputs.


## *Free lunch theorems*



By a No-Free-Lunch argument, no selector, learned or fixed, can dominate across all possible distributions of (dataset, partition, quality) triples; averaged over all problems, every selector ties. MetaIVM's gains must therefore rest on structural assumptions about the distribution of clustering problems actually encountered. The paper acknowledges this informally, but it never states the operative assumption as an assumption.

We see at least three candidate conditions that the paper's empirical results already gesture at but do not formalize:

1. A distributional assumption: deployment problems are drawn from, or lie within the support of, the training feature distribution.
2. A learnability/regularity condition on the map from features to external agreement
3. A feature-sufficiency condition: that the feature space separates good from bad partitions.

We do not expect a tight theoretical characterization of when learned clustering evaluation works. Our request is more modest: state the meta-distributional assumption explicitly (even informally), connect it to the No-Free-Lunch framing so the reader understands what is being traded for the performance gains, and ideally re-frame the transfer-asymmetry experiments as a diagnostic for that assumption rather than an isolated robustness check.

---

### Review · Reviewer_g9yy · 2026-07-11

**Summary Of Contributions:**

The paper proposes **MetaIVM**, a learned score for selecting clustering models when ground-truth labels are unavailable. During offline training, the method uses labeled datasets to learn how well a candidate clustering agrees with the reference labels. At deployment, it predicts this agreement using only label-free properties of the dataset and partition, such as cluster sizes, within-cluster variation, centroid separation, and graph connectivity. Because it scores individual partitions, it can select the clustering algorithm, its hyperparameters, and the number of clusters within one framework.

The evaluation contains 223 datasets and 16,889 candidate clusterings. The main model reduces selection regret from 0.217 for Calinski-Harabasz, the strongest classical baseline, to 0.071. The paper also includes feature ablations, candidate-pool experiments, cross-domain transfer, failure analysis, and a preliminary extension to graph community detection.

The main strengths are the important practical problem, the use of dataset-level train-test splits, the broad benchmark, and the careful effort to determine where the improvement comes from. In particular, the result that even a linear model using partition features outperforms classical validity measures is interesting. The main weaknesses are incomplete positioning relative to previous learned clustering-quality surrogates, missing comparisons with the closest methods, several inconsistencies in the reported experimental setup, and an imprecise theoretical proposition. The learned score also represents agreement with the benchmark labels, rather than a universal definition of clustering quality.

**Audience:**

Yes

**Audience Explanation:**

Selecting a clustering model without labels is a common and difficult problem. The paper provides evidence that features describing the actual candidate partition can be more useful than a fixed geometric validity measure. The analyses of partition features, graph features, difficult datasets, and cross-domain transfer should be relevant to researchers working on clustering, AutoML, meta-learning, and unsupervised evaluation.

The paper also provides a useful broader finding: the improvement appears to come mainly from the per-partition representation and training formulation, rather than from using a particularly complex prediction model. This is informative even if related learned-surrogate approaches already exist.

**Broader Impact Concerns:**

None.

**Claims And Evidence:**

No

**Claims Explanation:**

The central empirical finding is supported by substantial evidence. Across the benchmark, the learned models generally outperform Silhouette, Calinski-Harabasz, and Davies-Bouldin. The ablations also provide useful evidence that the improvement is not simply caused by applying XGBoost to existing validity scores. The dataset-level splits are appropriate and reduce the risk of direct information leakage.

However, I cannot answer "Yes" in the paper's current form for the following reasons.

First, the novelty discussion is incomplete. The paper repeatedly contrasts MetaIVM with previous work that only recommends an algorithm at the dataset level. However, AutoClust [1] already trains a regression model that maps internal validity measures of a candidate clustering to predicted external agreement. PoAC [2] also trains a surrogate on earlier labeled clustering problems and uses it to predict the quality of candidate clustering pipelines. These methods are not identical to MetaIVM, but they are close to its central formulation and should be discussed and compared directly. TMLR does not require every method to be completely novel, but claims about what has or has not been done previously must be accurate.

Second, the baseline comparison is incomplete. The main experiments contain only three classical validity measures and adjusted Calinski-Harabasz. This is a limited comparison for a paper about general clustering model selection, especially because the candidate pool includes density-based methods, non-spherical clusters, and partitions with noise. The paper should compare with stronger specialized validity measures and with learned-surrogate baselines based on AutoClust and PoAC.

Third, several parts of the experimental description are inconsistent. Table 1 reports a Ridge regret of 0.088 using 28 features, while Tables 2 and 20 report 0.123 for what appears to be the same setting. Section 5.2 and Table 9 list different hyperparameter grids for KMeans, DBSCAN, and HDBSCAN. The paper refers to six clustering algorithm families even though Spectral Clustering was skipped, apparently leaving five active families. The real-graph experiment also contains multiple graph-only settings whose relationship is not clearly explained. These inconsistencies make it difficult to identify the exact configuration supporting each headline result.

Fourth, the theoretical proposition needs correction. The paper states that being non-decreasing in between-cluster dispersion and non-increasing in within-cluster dispersion is equivalent to being a monotone function of their ratio. These conditions are not equivalent. The claim that the defined class includes Davies-Bouldin is also questionable because Davies-Bouldin depends on cluster-level quantities and is not determined only by the global within-cluster and between-cluster dispersion values used in the definition.

These issues do not necessarily invalidate the core empirical result, but they prevent the current set of claims from being fully supported in an accurate and clear manner.

## References

[1] Y. Poulakis, C. Doulkeridis, and D. Kyriazis. *AutoClust: A Framework for Automated Clustering Based on Cluster Validity Indices*. IEEE International Conference on Data Mining, 2020.

[2] M. Camilo da Silva, G. Marques Tavares, E. Medvet, and S. Barbon Junior. *Problem-oriented AutoML in Clustering*. arXiv:2409.16218, 2024.

**Requested Changes:**

### Critical to securing my recommendation for acceptance

1. **Correct and expand the related-work discussion.**

   Discuss AutoClust [1], PoAC [2], and any other methods that train learned surrogates for external clustering quality. Clearly explain how MetaIVM differs from these methods. The distinction may lie in its richer partition representation, graph features, direct scoring of arbitrary candidate partitions, larger benchmark, or stronger evaluation, but the paper should not imply that learned per-candidate quality prediction is entirely new.

2. **Compare against the closest learned methods.**

   Add an AutoClust-style baseline that predicts external quality from a broad collection of internal validity measures. Also include a PoAC-style surrogate if feasible. These baselines should use the same datasets, candidate partitions, train-test splits, and evaluation metrics as MetaIVM. At minimum, the authors should reproduce the central surrogate components of these methods rather than comparing only with dataset-level algorithm selectors.

3. **Resolve the numerical and experimental inconsistencies.**

   Reconcile the Ridge results in Tables 1, 2, and 20. Make the clustering configurations and hyperparameter grids consistent between Section 5.2 and Table 9. State clearly how many algorithm families actually produced candidate partitions. Clarify the different graph-only training settings in Table 16. A single table listing the exact feature set, training objective, split, and model configuration behind each headline number would improve clarity.

4. **Correct or narrow the theoretical proposition.**

   Provide a mathematically precise definition of the class of validity measures covered by the result. Remove the claimed equivalence between separate monotonicity in \(B\) and \(W\) and monotonicity in \(B/W\). Do not claim that Davies-Bouldin or Silhouette belongs to the defined class unless this is formally established. If the result is intended only as an illustration rather than a general theorem, it should be presented as such.

5. **Strengthen the baseline evaluation.**

   Include representative validity measures designed for density-based clustering, graph structure, non-convex clusters, and partitions with noise. Explain how degenerate partitions, singleton clusters, and noise assignments are handled for every baseline. The paper already mentions more specialized measures in the related-work section, but these are not included in the main empirical comparison.

6. **Clarify validation and early stopping.**

   Confirm that all model selection, feature selection, early stopping, and hyperparameter tuning preserve dataset-level separation. The MLP description mentions a built-in validation-fraction mechanism, which could split candidate partitions from the same dataset unless it was replaced with a dataset-level validation procedure.

### Changes that would strengthen the work

1. Add leave-one-generator-family-out or leave-one-geometry-out experiments for synthetic datasets. Randomly dividing datasets produced by closely related generators may overestimate generalization.

2. Report the main results separately for synthetic, OpenML, text, and image datasets, including uncertainty estimates. The real-data-only result is especially important for practical relevance.

3. Evaluate how performance changes under controlled shifts in cluster count, dimensionality, noise level, imbalance, and cluster shape.

4. Discuss more clearly when reference class labels are an appropriate clustering target. A clustering that agrees with class labels is not necessarily the most useful partition for every application.